# Effects of Different Fertilizers on Soil Microbial Diversity during Long-Term Fertilization of a Corn Field in Shanghai, China

**Chenyan Sha †, Jian Wu †, Jianqiang Wu, Chunmei Ye, Cheng Shen, Jinghua Su** and **Min Wang \***

Shanghai Academy of Environmental Sciences, Shanghai 200235, China
* Correspondence: saeswangm78@163.com
† These authors contributed equally to this work.

**Abstract:** The long-term applications of different fertilizers (chicken manure, swine manure, and organic fertilizer) on the microorganisms of a corn field were investigated. The microbial communities during four periods (seedling, three-leaf, filling and mature periods) were comprehensively studied with molecular biology technology. Results showed that most nutrient contents (organic matter, nitrogen, phosphorus, and potassium) and levels of several heavy metals (As, Pb, and Cr) in the chicken and swine manures were higher than those in the organic fertilizer. The alpha diversity varied during the long-term fertilization, and the chicken manure was the best fertilizer to maintain the abundance of microorganisms. The microbial community of soil changes over time, regardless of the addition of different fertilizers. The correlations between environmental factors and microbial communities revealed that nutrient substances (available nitrogen, available potassium, and $NO_3$-N) were the most significant characteristics with the chicken and swine manures, while organic matter and nitrogen exhibited similar effects on the microbial structure with the organic fertilizer. The Pearson correlations of environmental factors on genus were significantly different in the organic fertilizer tests compared with the others, and *Pseudomonas*, *Methyloligellaceae*, *Flavobacterium*, and *Bacillus* showed significant correlations with the organic matter. This study will provide a theoretical basis for improving land productivity and sustainable development in corn fields.

**Keywords:** different fertilizers; long-term fertilization; microbial community; environmental factors; correlation



## 1. Introduction

The application of manure as fertilizer to agricultural land is a common practice around the world [1]. Livestock and poultry breeding is the pillar industry of China's agriculture. In order to ensure the effective supply of livestock and poultry products, this industry has been developing intensively and on a large scale in recent years. The treatment and disposal of a large amount of livestock and poultry manure produced by large-scale breeding is an effective way to reduce livestock and poultry waste and to recycle resources. Microorganisms, nutrients, heavy metals, and antibiotics change the soil environment after livestock application, while the biomass, community diversity and composition, and functional flora of soil microorganisms are affected as well [2].

Long-term or short-term livestock and poultry manure application or combined application with straws and chemical fertilizers could affect the physical and chemical characteristics of the soil, thereby significantly affecting the community structure, abundance, and diversity of farmland soil microorganisms. Qin et al. [3] found that long-term fertilization has a significant impact on the bacterial community structure of black soil. Giacometti et al. [4] also found that under a condition of long-term fertilization, the bacterial abundance of farmland soil increased significantly with the increase in swine manure application, while its community diversity significantly decreased. Xun et al. [5] showed that the short-term

fertilization of swine manure improved soil nutrient content and had a significant impact on soil microbial diversity. Li et al. [6] revealed that the application of chicken and swine manure can effectively increase the diversity and abundance of soil bacterial communities in farmland. The changes in amino-acid carbon sources and carbohydrates in the soil were the main reasons that affected the soil microbial diversity changes. Hence, improving soil microbial diversity helps enhance the stability of farmland ecosystem functions and their resistance and resilience against environmental disturbances.

The changes in the microbial community after livestock manure application have been widely studied. It was reported that an increase in soil nutrient content and the introduction of exogenous bacteria affect the growth and reproduction of soil microorganisms. An increase in soil nutrient content enhances the growth and reproduction of eutrophic microorganisms in farmland soil, while it reduces the abundance of oligotrophic flora [7,8]. Adding organic or inorganic fertilizers can reduce the relative abundance of oligotrophic Acidobacteriain black and farmland soil in Northeast China [9]. However, in one wheat-corn rotation system, the application of swine manure and straw or the combined application of swine manure fertilizers inhibited the growth of soil dominant bacteria, such as Bacteroidetes, Acidobacteria, and Gemmatimonadetes, in a short period [10]. Changes in soil nitrogen content have a significant impact on soil high-abundance bacteria. Short-term nitrogen application may reduce the abundance of Actinobacteriaand Nitrospirae in farmland soil [11], while long-term nitrogen application is beneficial for the deformation of the growth of Proteobacteria and Actinobacteria [12]. Actinomycetes are more sensitive to the addition of exogenous nutrients. Meanwhile, untreated livestock and poultry manure contain a large number of pathogenic microorganisms, such as Salmonella, *E. coli*, Mycobacterium tuberculosis, etc. Most of them can survive in the soil for a long time after entering the farmland, and some pathogenic bacteria such as *E. coli* can continue to survive and reproduce after entering the soil. It is easy for them to enter the human body through the food chain and cause food-borne diseases, which seriously threatens the health of the soil environment [13]. However, comparisons between the microbial communities with livestock manure and those with organic fertilizer in the long-term fertilization of corn were rarely considered and need to be further studied.

In the applications of different fertilizers, the nutrients and pollutants have greater impacts on the functional microbes and functional flora of the soil. Accordingly, the correlations between environmental factors and microorganisms should be studied. When the nutrients in fertilizers enter the soil, they affect the soil nutrient cycle and the decomposition of organic matter [6]. Changes in soil nutrients after the application of manure also have a great impact on the functional plants involved in the turnover of soil organic matter or the utilization of carbon sources. Li et al. [14] studied the effects of adding livestock manure on soil functional microorganisms and found that the application of chicken and swine manure can effectively increase the carbon source utilization capacity of the soil microbial community by changing the soil nutrients in the farmland. Guo et al. [15] evaluated the effect of swine manure on the carbon conversion of soil during long-term application, finding that it increased the diversity of soil microorganisms and improved the soil microbial organic matter turnover capacity and carbon source utilization rate. In addition, after pollutants such as heavy metals and antibiotics in livestock manure enter the soil, they can also reduce the abundance of ammonia-oxidizing bacteria, inhibit the nitrification and denitrification of soil microorganisms, and reduce the utilization of carbon sources by microorganisms, thereby affecting the biogeochemical cycles of nitrogen and carbon in the soil [16]. Therefore, the comparison of nutrients, heavy metals, and their correlations with microbial communities between different fertilizers was necessarily investigated.

In this study, different fertilizers such as chicken manure, swine manure, and organic fertilizer were applied in the long-term fertilization of corn. The physical and chemical characteristics and heavy metals of different fertilizers were analyzed. Moreover, the diversity and richness of the microbial communities as well as the microbial compositions and structures were comprehensively compared. In particular, the correlations between envi-

ronmental factors and microbial communities with the applications of different fertilizers were further investigated. This study will provide a theoretical basis for improving land productivity and sustainable development in corn fields.

## 2. Materials and Methods

### 2.1. Experimental Location

This research experiment was carried out in Hongxing Village, Zhongxing Town, Chongming Island (121°09′–121°54′, 31°27′–31°51′ N). This area is located at the mouth of the Yangtze River and belongs to the northern subtropical region. The climate is mild and humid. The annual average sunshine number is 2094.2 h, the annual average temperature is 15.3 °C, the annual average rainfall is 1025 mm, and the relative air humidity is above 80%. The natural geographical condition of the region is suitable for the development of the planting industry. The experimental base was established in 2010 and grows representative crops in the Chongming District, mainly yellow corn and cauliflower. The plot selected for this experiment covers an area of 300 m$^2$, mainly corn planting plots.

### 2.2. Experimental Set-Up

Chicken manure (CM), swine manure (SM), and organic fertilizer (OF) were selected as the experimental manures. The chicken manure was collected from a chicken farm, while the swine manure and organic fertilizer were collected from a swine farm in Chongming. The organic fertilizer was composted under aerobic conditions from swine manure, mushroom, and straw wastes.

As a common crop in Chongming, yellow corn was selected as the planting crop in this experiment. Direct sowing was conducted in all the experiments. The sowing time was early April in 2020. The planting density was 52,500 plants per mu. Other cultivation and management measures were the same as those in the field.

Ten treatments including nine fertilization treatments and one control treatment (CK) were designed. Each treatment had a repetition of 3 times. The concentration gradient of the manure application was set in accordance with the standard procedures in "Technical Specifications for Returning Livestock and Poultry Manure to Field (GB/T 25246-2010)", which were 2 kg·m$^{-2}$, 4 kg·m$^{-2}$, and 6 kg·m$^{-2}$. Different concentrations of the chicken manure, swine manure, and organic fertilizer were applied as base fertilizers to the soil at one time before sowing the corn and then plowed after spreading fertilization.

Thirty experimental plots (3 m × 3 m) were set up, and a 0.5 mm thickness impermeable membrane was used to block the plots. The buried depth of the impermeable membrane was 30 cm to prevent mutual interference between the plots. At the same time, a channel for on-site monitoring and sample collection was isolated between each plot, with a width of about 20 cm. The layout of the experimental plots is shown in Figure 1.

### 2.3. Soil Sample Collection

The soil samples were collected in late April (seedling period), mid-May (three-leaf period), early July (filling period), and late August (mature period) in 2020.According to the "S" sampling method, five points were randomly selected from each sample plot and rhizosphere soil samples were collected from each of the 5 points by using a sterile stainless-steel soil drill (0–20 cm depth). Finally, the samples of five points were mixed evenly. A PVC soil respiration ring base (soil collar) with a diameter of 20 cm and a height of 12 cm was pre-embedded in each parallel plot on the site. After the corn was sowed, it was embedded in the soil with a depth of 10 cm and the uppercut was 2 cm above the ground. In order to avoid experimental errors, the roots of the corn plants should be avoided when the base is embedded and placed in the gaps between the plants. In order to eliminate the influence of green plant photosynthesis and surface litter on the measurement, the day before each measurement, the plants and litter in the ring should be removed, and the surface soil should not be disturbed. Soil samples of 500 g from each plot were collected. After removing debris, earthworms, and plant residues in each soil sample,

each soil sample was divided into two portions: the first portion was stored at 4 °C for the analysis of physical and chemical characteristics, whereas the other was stored at −80 °C for DNA extraction and microbial analysis.

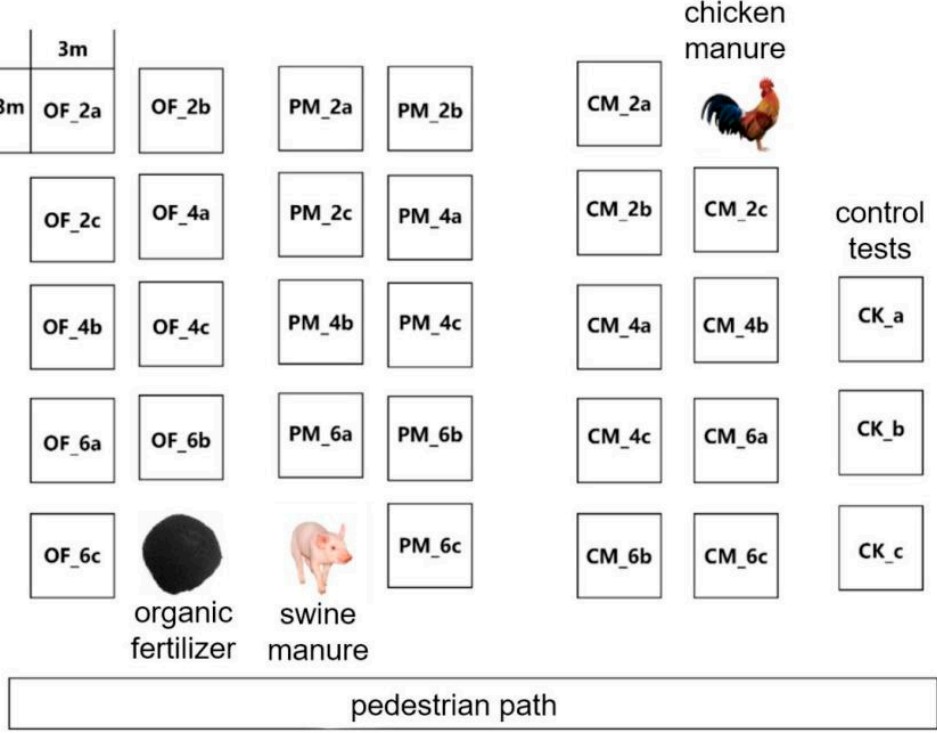

**Figure 1.** The experimental site layout (CK: control; CM: chicken manure; OF: organic fertilizer; SM: swine manure;PM represents swine manure).

*2.4. Analysis Methods*

2.4.1. Physical and Chemical Characteristics

The samples brought back to the laboratory were placed in the room for natural air drying, and, after grinding, they were passed through 10-mesh and 100-mesh sieves and placed in a refrigerator at 4 °C for further analysis. Organic matter (OM) and organic carbon (OC) were measured using a total organic carbon analyzer (TOC-VCPH, Shimadzu, Japan). Total nitrogen (TN), total phosphorus (TP), total potassium (TK), available nitrogen (AN), available phosphorus (AP), and available potassium (AK) were measured according to the Standard Methods [17]. Metal ion concentrations such as lead (Pb), cadmium (Cd), arsenic (As), mercury (Hg), and Chromium (Cr) were determined using an inductively coupled plasma emission spectrometer (ICP-AES, Agilent, Santa Clara, CA, USA).

2.4.2. Microbial Community Characterization

Microbial DNA was extracted from sludge using the E.Z.N.A.® Soil DNA kit (Omega Bio-Tek, Norcross, GA, USA) according to the manufacturer's protocols and then pooled together. DNA extracts were checked on 1% agarose gel, and DNA concentration and purity were determined with a Nano Drop 2000 UV-vis spectrophotometer ( Thermo Scientific, Wilmington, NC, USA). For the bacterial community, the bacterial 16S rRNA genes were amplified using the universal bacterial primers 27F (5′-AGRGTTYGATYMTGGCTCAG-3′) and 1492R (5′-RGYTACCTTGTTACGACTT-3′). The primers were tailed with PacBio barcode sequences to distinguish each sample. The amplification reactions (20 μL volume) consisted of $5 \times 4$ μL of FastPfu buffer, 2 μL of 2.5 mM dNTPs, 0.8 μL of forward primer (5 μM), 0.8 μL of reverse primer (5 μM), 0.4 μL of FastPfu DNA Polymerase, 10 ng of template DNA, and DNase-free water. The PCR amplification was performed as follows: initial denaturation at 95 °C for 3 min, followed by 27 cycles of denaturing at 95 °C for

30 s, annealing at 60 °C for 30 s and extension at 72 °C for 45 s, single extension at 72 °C for 10 min, and end at 4 °C (ABI Gene Amp® 9700 PCR Thermocyclerm, Santa Clara, CA, USA). The PCR reactions were performed in triplicate. After electrophoresis, the PCR products were purified using the AM Pure® PB beads (Pacifc Biosciences, Menlo Park, CA, USA) and quantified with a Quantus™ Fluorometer (Promega, Fitchburg, WI, USA). The optimized CCS reads were clustered into operational taxonomic units (OTUs) using UPARSE 7.1 with a 97% sequence similarity level. The most abundant sequence for each OTU was selected as a representative sequence. To minimize the effects of sequencing depth on alpha and beta diversity measures, the number of 16S rRNA gene sequences from each sample was rarefied to 6000, which still yielded an average Good's coverage of 99.09%. The alpha diversity indices were calculated in MOTHUR (http://www.mothur.org (accessed on 9 September 2020)). The Pearson correlation coefficient was calculated to evaluate the correlations between environmental factors and individual microorganisms according to Ping et al. [18].

### 2.5. Statistical Analysis

Excel 2019, SPSS 24, and Origin 2017 were used for the data analysis and graphing. The graphs in Figure 2 were built with Origin. The differences between the different treatment groups were compared using the one-way analysis of variance method, while the multiple comparison was performed using the LSD (Least Significance Difference) method ($p < 0.05$).

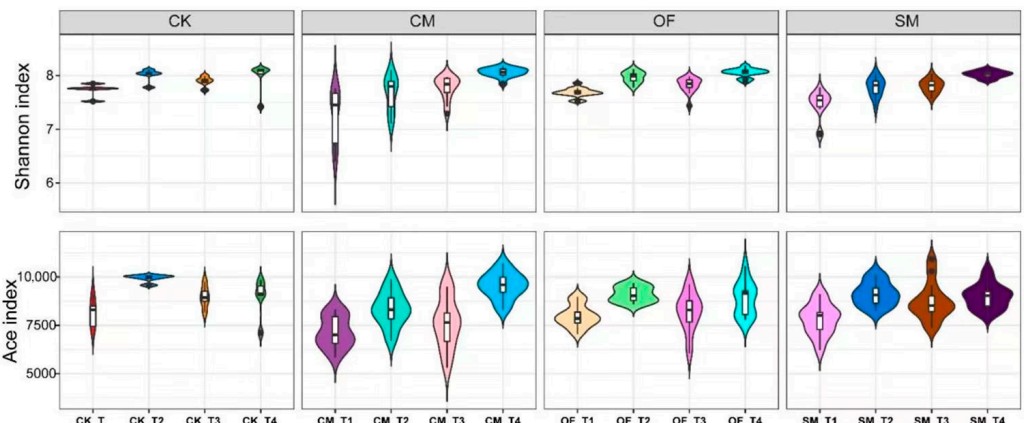

**Figure 2.** The Shannon and Ace indices of microbial community with different fertilizers (CK: control; CM: chicken manure; OF: organic fertilizer; SM: swine manure).

## 3. Results and Discussion

### 3.1. Analysis of Physical and Chemical Characteristics

The physical and chemical characteristics of the different fertilizers are shown in Table 1. The pH value of the organic fertilizer was the highest (7.42), and was significantly higher than that of CM and SM ($p < 0.05$). The contents of OM, TN, AN, TK, and AK in CM were the highest among the three fertilizers. The concentrations of TK and AK were significantly higher than those in SM ($p < 0.05$), measuring 81.95% and 118.53% higher, respectively. The highest contents of TP and AP appeared in the swine manure, and its TP concentration was higher than that of CM ($p < 0.05$). OF had the lowest nutrient content, and its OM, TN, TP, TK, AN, AP, and AK contents were significantly lower than those of CM ($p < 0.05$), which were reduced by 34.71%, 37.05%, 45.04%, 1.86%, 80.99%, and 41.45%, respectively. Similarly, the OM, TN, TP, AN, and AP contents of OF were significantly lower than those of SM ($p < 0.05$), which were reduced by 43.38%, 32.91%, 57.50%, and 82.28%, respectively. It was reported that fresh swine manure usually has more soluble phosphorus [19], which is in accordance with this study in which the concentrations of TP and AP in SM were higher than those in CM and OF. Moreover, the chicken manure had the highest nutrient content amongst the livestock manure, which was similar to the results

of Ksheem et al. [20]. The nutrient content of OF is relatively low, which may relate to the high temperature during the composting process [21,22].

**Table 1.** The characteristics of chicken manure, swine manure, and organic fertilizer.

| Characteristics | Chicken Manure | Swine Manure | Organic Fertilizer |
|---|---|---|---|
| pH | $6.31 \pm 1.23$ [b] | $6.60 \pm 0.78$ [b] | $7.42 \pm 0.93$ [a] |
| Moisture (%) | $64.2 \pm 9.28$ [a] | $56.9 \pm 6.01$ [b] | $40.8 \pm 4.79$ [c] |
| OM (g·kg$^{-1}$) | $844 \pm 39.92$ [a] | $790 \pm 18.80$ [a] | $551 \pm 25.61$ [b] |
| TN (g·kg$^{-1}$) | $34.3 \pm 8.23$ [a] | $32.2 \pm 3.08$ [a] | $21.6 \pm 2.46$ [b] |
| AN (mg·kg$^{-1}$) | $3.17 \pm 0.47$ [a] | $2.73 \pm 0.22$ [a] | $1.16 \pm 0.23$ [b] |
| TP (g·kg$^{-1}$) | $18.2 \pm 0.23$ [b] | $29.10 \pm 4.22$ [a] | $12.60 \pm 1.27$ [c] |
| AP(mg·kg$^{-1}$) | $3.02 \pm 0.18$ [a] | $3.24 \pm 0.35$ [a] | $0.574 \pm 0.09$ [b] |
| TK (g·kg$^{-1}$) | $3.73 \pm 0.55$ [a] | $2.05 \pm 0.39$ [b] | $2.05 \pm 0.11$ [b] |
| AK (mg·kg$^{-1}$) | $21.7 \pm 5.43$ [a] | $9.93 \pm 1.91$ [b] | $12.7 \pm 2.09$ [b] |

Note: Different letters (a, b, c) indicate that there were significant differences between different fertilizers with each characteristic ($p < 0.05$).

As can be seen in Table 2, the heavy metal contents of the three fertilizers did not exceed the heavy metal concentration limit (NY525-2012) [23]. The concentrations of Cd and Hg were not significantly different between the three fertilizers. The highest concentrations of As and Pb were observed in CM, measuring at 0.64 mg·kg$^{-1}$ and 19.2 mg·kg$^{-1}$, respectively. The As concentration in CM was 48.83% higher than that in SM ($p < 0.05$), while the Pb and As concentrations were significantly higher than those in OF ($p < 0.05$), measuring 28.00% and 357.14% higher, respectively. The As content in SM was also significantly higher than that in OF, measuring 207.12% higher. The Cr concentration (25 mg·kg$^{-1}$) was the highest in SM, followed by CM. The Cr concentrations in CM and SM were significantly higher than that in OF ($p < 0.05$), measuring 48.38% and 61.29% higher, respectively. The analysis results showed that the heavy metals As, Pb, and Cr in the chicken manure or SM were more abundant than in OF.

**Table 2.** Heavy metal concentrations in chicken manure, swine manure, and organic fertilizer.

| Heavy Metal | Chicken Manure | Swine Manure | Organic Fertilizer |
|---|---|---|---|
| Cd (mg·kg$^{-1}$) | $0.24 \pm 0.02$ [a] | $0.27 \pm 0.04$ [a] | $0.25 \pm 0.00$ [a] |
| Hg (mg·kg$^{-1}$) | $0.156 \pm 0.04$ [a] | $0.173 \pm 0.01$ [a] | $0.157 \pm 0.01$ [a] |
| As (mg·kg$^{-1}$) | $0.64 \pm 0.09$ [a] | $0.43 \pm 0.00$ [b] | $0.14 \pm 0.00$ [c] |
| Pb (mg·kg$^{-1}$) | $19.2 \pm 0.04$ [a] | $13.3 \pm 1.03$ [b] | $15.0 \pm 1.21$ [ab] |
| Cr (mg·kg$^{-1}$) | $23 \pm 0.97$ [a] | $25 \pm 2.73$ [a] | $15.5 \pm 3.88$ [b] |

Note: Different letters (a, b, c) indicate that there were significant differences between different fertilizers with each heavy metal.

### 3.2. Diversity and Richness of Microbial Communities in Long-Term Fertilization

In the microbial community analysis, the rarefaction curves (at a 97% sequence similarity) of all the samples plateaued, demonstrating that the number of pyrosequencing reads was enough to explain the large fraction of OTUs in the soil samples.

Alpha diversity indices can be used to quantitatively estimate different microbes in a dataset and determine the total number of microbes. The Shannon and Ace indices were selected to reflect the diversity and richness of soil microorganisms with different fertilization treatments (Figure 2). The Shannon and Ace indices in the OF and CK soils demonstrated no significant difference in different periods, indicating that the application of organic fertilizer has little effect on the diversity and abundance of soil microbial communities. For the Ace index, there was no significant difference between the SM and CK soils during the four periods, while the Shannon index of the SM soil was lower than that of the CK soil during the four periods and reached a significant level in the first three periods ($p < 0.05$), indicating that the swine manure reduced the diversity of the soil microbial

community. However, it had a small effect on the microbial abundance. The application of swine manure reduces the diversity of the soil microbial community, and the reduction in the microbial community diversity leads to a reduction in the efficiencies of the ecosystem functions [24]. The Shannon and Ace indices of the CM soil both increased with time, although they were significantly lower than CK in the first two periods ($p < 0.05$); however, by the mature stage, there was no significant difference between CM and CK. This reveals that the chicken manure greatly increased the diversity and richness of microorganisms during the four periods. In addition, the richness of CM in the last stage was significantly higher than that of OF and SM ($p < 0.05$), indicating that the chicken manure was the best fertilizer to maintain the abundance of microorganisms. Campbell et al. [25] also found that from a long-term perspective, the application of chicken manure is conducive to improving the diversity of soil microbial communities. An increase in microbial community diversity is conducive to enhancing the stability of a soil ecosystem, making it more resistant and able to recover in the face of environmental disturbances [26].

### 3.3. Microbial Composition and Structure in Long-Term Fertilization

The bacterial communities determined at the phylum and genus levels were further analyzed by comparing populations among the samples. A total of 42 phyla were detected, and the phyla with a relative abundance of less than 0.1% were merged into others (Figure 3a). It can be seen in Figure 3a that Proteobacteria was the most abundant phylum in all soils with different treatments, accounting for 27.95~39.84% of the total bacterial counts in each sample. This result was consistent with previous studies of soil microbial communities [27]. The main phyla present in these soil samples with different treatments were quite similar, including Actinobacteria, Acidobacteria, Acidobacteria, Chloroflexi, Bacteroidetes, Firmicutes, Gemmatimonadetes, Rokubacteria, Patescibacteria, Planctomycetes, Verrucomicrobia, Nitrospirae, and Latescibacteria, and their total abundance accounts for 97.86~98.68%.

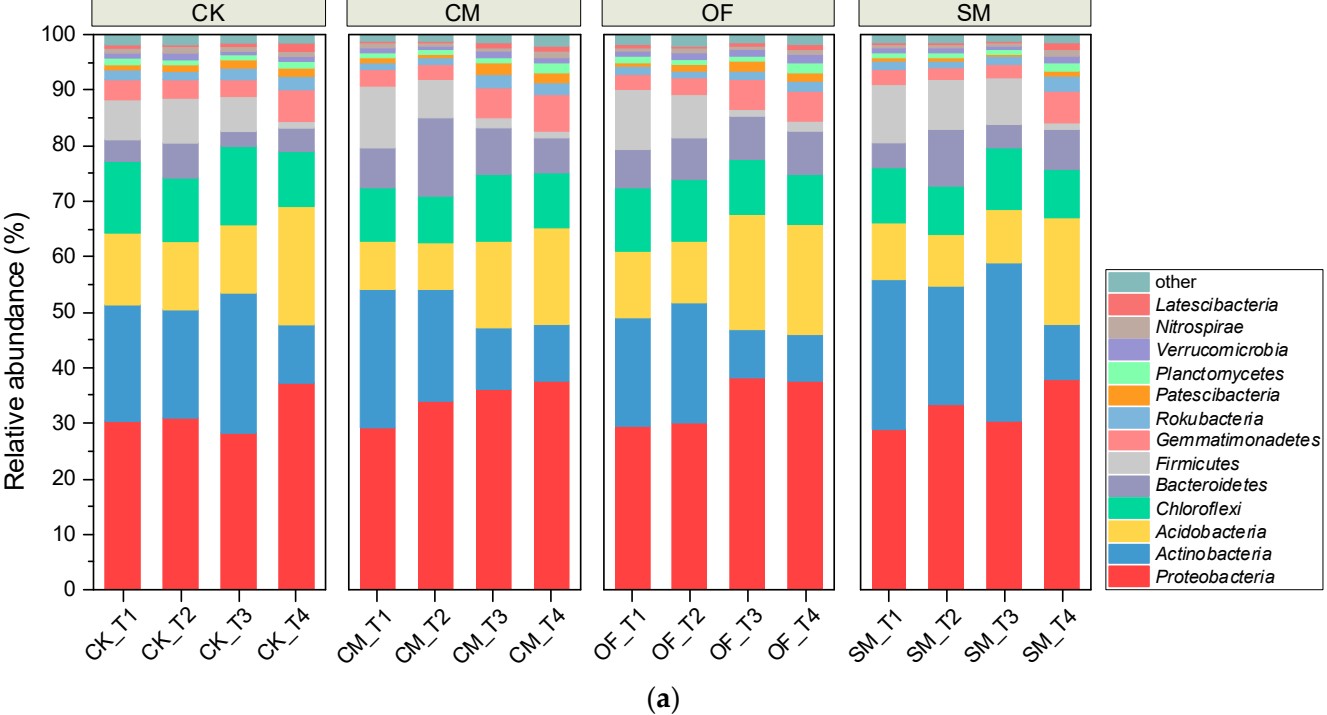

**(a)**

**Figure 3.** *Cont.*

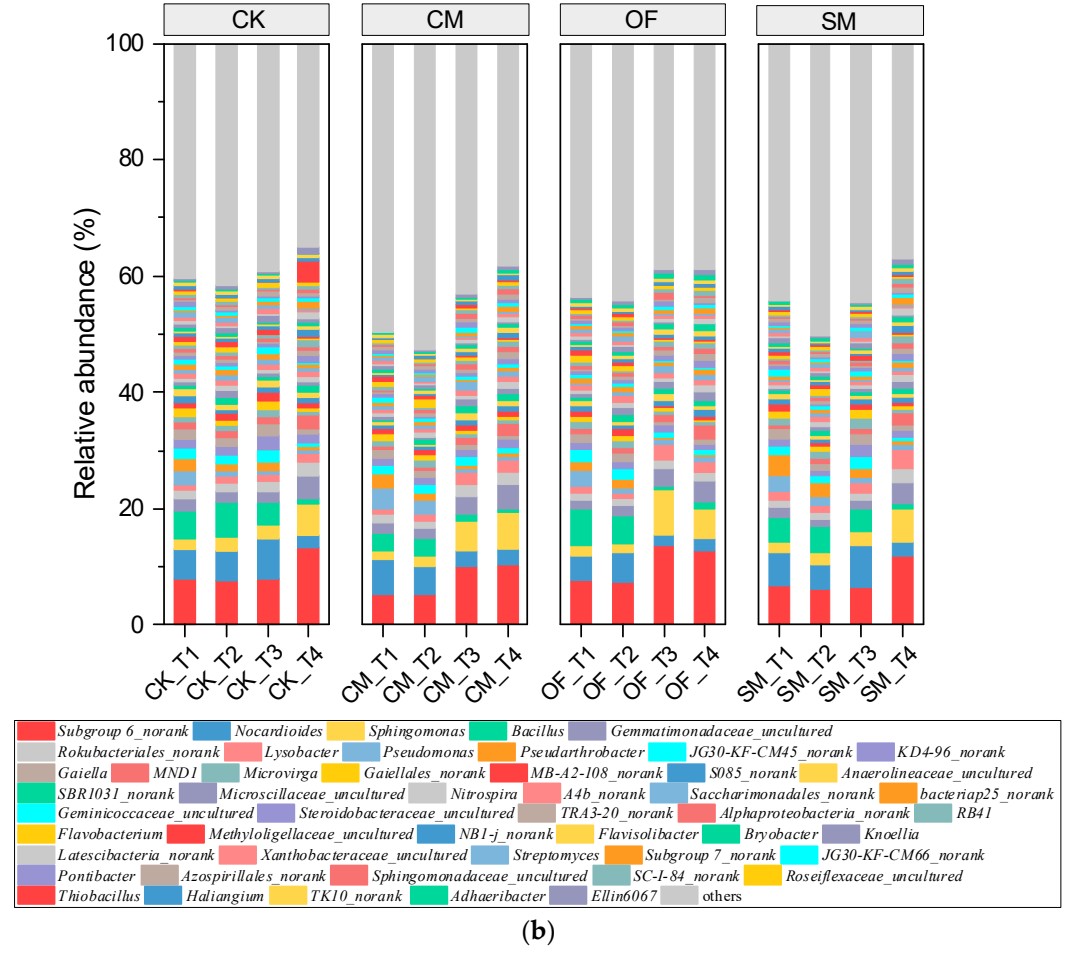

**Figure 3.** The relative abundances of microorganisms on phylum (**a**) and genus (**b**) levels with different fertilizers (CK: control; CM: chicken manure; OF: organic fertilizer; SM: swine manure; T1: seedling period; T2: three-leaf period; T3: filling period; T4: mature period).

There were certain differences in the second dominant phyla of the soil in each treatment group. At the seedling and three-leaf stage, the second dominant phylum of each treatment group was Actinobacteria, accounting for 19.49–26.91% of the total bacterial abundance. During the filling stage, the second dominant phylum was still Actinobacteriain the CK and SM tests, accounting for 25.26% and 25.84%, respectively. However, the abundance of Acidobacteria was greatly increased in the CM and OF tests, measuring at 14.21% and 19.62%, respectively. It was reported that Acidobacteriais one of the dominant bacteria in soil, and it is easy to accumulate in acid soil [28]. The decrease in soil pH after the application of chicken manure and organic fertilizer may be the main reason for the increased abundance in Acidobacteria. In the mature period, Acidobacteriabecame the second dominant bacteria phylum in the soils of all the treatments, accounting for 16.30–20.48%. From the changes in the dominant microbial phyla in different periods, it can be found that the microbial community of soil changes over time regardless of the addition of different manure, which is consistent with the results of previous studies [29]. The changes in microorganisms on the genus level were similar to those on the phylum level. It can be seen in Figure 3b that the relative abundances of *Subgroup 6_norank*, *Sphingomonas*, *gemmatinonadaceae_uncultured*, and *Lysobacter* were greatly increased in the mature period, while *Nocardioides* and *Bacillus* were significantly inhibited.

Furthermore, the one-way analysis of variance was selected to analyze the differences in the compositions of microorganisms between treatment groups at different periods based on the relative abundances of phylum-level microorganisms. The effects of fertil-

ization on the microbes in the soil with high abundance levels are highly consistent. The application of CM, OF, and SM all increased the relative abundance of Firmicutes and Bacteroidetes. Compared with CK, the relative abundance of Firmicutes in the soil of CM, OF, and SM at the seedling stage was significantly higher ($p < 0.05$), measuring at 68.45%, 70.03%, and 49.77% higher, respectively. It was known that Firmicutes can be enriched at a rapid multiplication rate in a growth environment where soluble organic matter and nutrient concentrations are high [30]. Therefore, it was significantly enhanced with the application of different fertilizers. During the entire growth period, the relative abundance of Bacteroidetes in the fertilized soil was higher than that of CK. Bacteroidetes reached a significant level in the seedling, filling, and mature stages ($p < 0.05$) with the OF fertilization, and they reached a significant level in all periods ($p < 0.05$) in the CM tests. Similarly, the relative abundance was significantly higher in the CK tests after the three-leaf stage ($p < 0.05$). During the entire growth period, the relative abundances of Bacteroidetes with the CM, OF, and SM fertilizations were higher than that in CK by 122.50%, 72.45%, and 55.33%, respectively. Bacteroidetes are eutrophic bacteria, and the increase in soil nutrients after manure application is the main reason for the increase in their abundance [8].

The analysis of the relative abundance of bacteria can only show the affected phyla and genera after fertilization, while the influence of fertilization on the bacterial structure is difficult to judge. Therefore, a principal component analysis (PCA) was used to study the similarities and differences in the microbial community structures between different fertilization treatment tests on the OTU level [31]. It was apparent that the four types of treatments during four periods can be divided into two categories (Figure 4). During the corn filling period, the samples of different treatment groups were clearly distinguished, indicating that the microbial community structures of the different treatment groups were quite different at this time. Among them, the SM and CK sample points are closer, and the OF and CM sample points are closer, indicating that the microbial community structures of the SM and CK samples were quite similar during this period. However, the microorganisms with OF are more similar to those with CM. This difference may be related to the nutrient content of the soil. At the maturity stage, the sample points of the different treatment groups are distinguished to a certain extent, but they are not clearly separated, indicating the microbial community structures between the different groups were relatively similar at this time. That was because the microorganisms continued to change over time, and they became stable during the mature period, indicating that the microbial structures were relatively sensitive to the change in temperature.

### 3.4. Correlations between Environmental Factors and Microbial Communities

In order to determine the interaction between environmental factors and microbial communities, the RDA redundancy and Pearson correlation coefficient were used to evaluate the effects of external factors on the microorganisms.

The RDA redundancy analysis showed that heavy metals such as, Cd, and Cr were more influential than organic characteristics (OM, OC) in the control tests. However, in the treatments with CM and SM, nutrient substances such as AN, AK, and $NO_3$-N became the most significant characteristics (Figure 5). That was because the nutrient composition, especially nitrogen, was abundant in the livestock manures [32,33]. An increase in nitrogen provides abundant substrates for microorganisms and stimulates the growth and reproduction of soil bacteria, which has a greater impact on the community structure. In addition, As was also a greatly affected factor in the CM and SM tests since the concentrations of As in CM and SM were significantly higher than that in OF. In the OF tests, organic (OM and OC) and nutrient characteristics (AN and $NO_3$-N)exhibited similar effects on the microbial structure, and the effect of heavy metals was relatively less than in the other tests, which was in accordance with the characteristics of OF in the previous section.

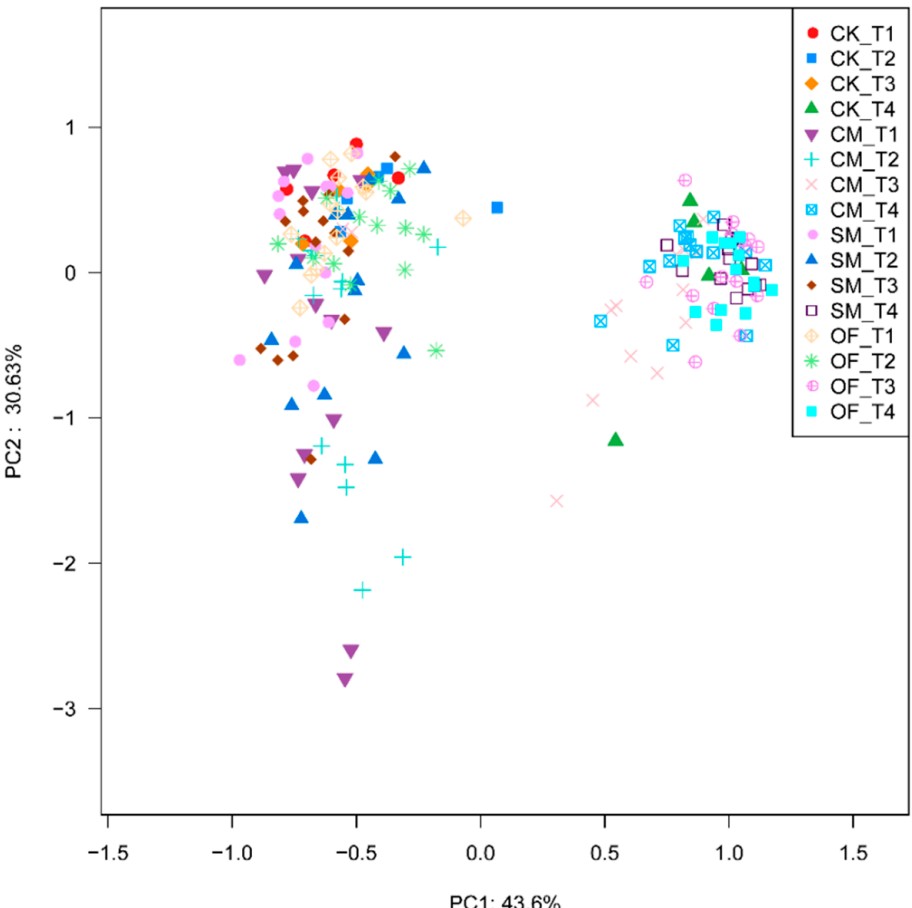

**Figure 4.** The PCA analysis of all treatment samples (CK: control; CM: chicken manure; OF: organic fertilizer; SM: swine manure; T1: seedling period; T2: three-leaf period; T3: filling period; T4: mature period).

A Pearson correlation analysis can further verify the correlations between soil environmental factors and dominant bacteria on the genus level. More significant correlation coefficients were observed with the fertilizer treatments than with the control tests (Figure 6), indicating that the addition of the fertilizers changed the environmental conditions of the soil and produced a great variety of microorganisms. It was observed that $NO_3$-N, AN, TP, and AK significantly correlated with most genera in the CM tests (Figure 6a). *Sphingomonadaceae*, *Saccharimonadales*, *Microscillaceae*, *Lysobacter*, *Chitinophagaceae*, and *Adhaeribacter* all exhibited positive correlations with TP, AN, and AK, while *Roseiflexaceae*, *Pseudarthrobacter*, *Nocardioides*, *Methyloligellaceae*, *Knoellia*, *Ilumatobacteracear*, *Geminicoccaceae*, *Gaiellales*, *Gaiella*, and *Bacillus* exhibited negative correlations with them. A similar result was observed in the SM tests, such that $NO_3$-N, AN, and AK were the most significant factors on genus (Figure 6d). The Pearson correlations of environmental factors on genus in the OF tests were significantly different compared with the other tests. It can be seen in Figure 6b that OM and OC exhibited the maximum number of significant co-efficiencies, revealing that organic compounds were the most significant factor in the OF tests. *Pseudomonas*, *Methyloligellaceae*, *Flavobacterium*, and *Bacillus* exhibited significant correlations with the organic matter; therefore, it can be deduced that they could rapidly grow and reproduce with high levels of organic matter in the soil. It was indicated that members of the genus *Pseudomonas* are a ubiquitous component of soil and rhizospheric ecosystems, where they play multifarious roles, such as in the recycling of organic matter [34]. *Methylotrophic* bacteria play a significant role in the biogeochemical cycle, and they are especially involved in phosphorous, nitrogen, and carbon cyclingin soil ecosystems [35,36]. Pijanowsk et al. [37]

revealed that *Pseudomonas* and *Bacillus* had the highest hydrocarbon biodegradation rates in short-term experiments.

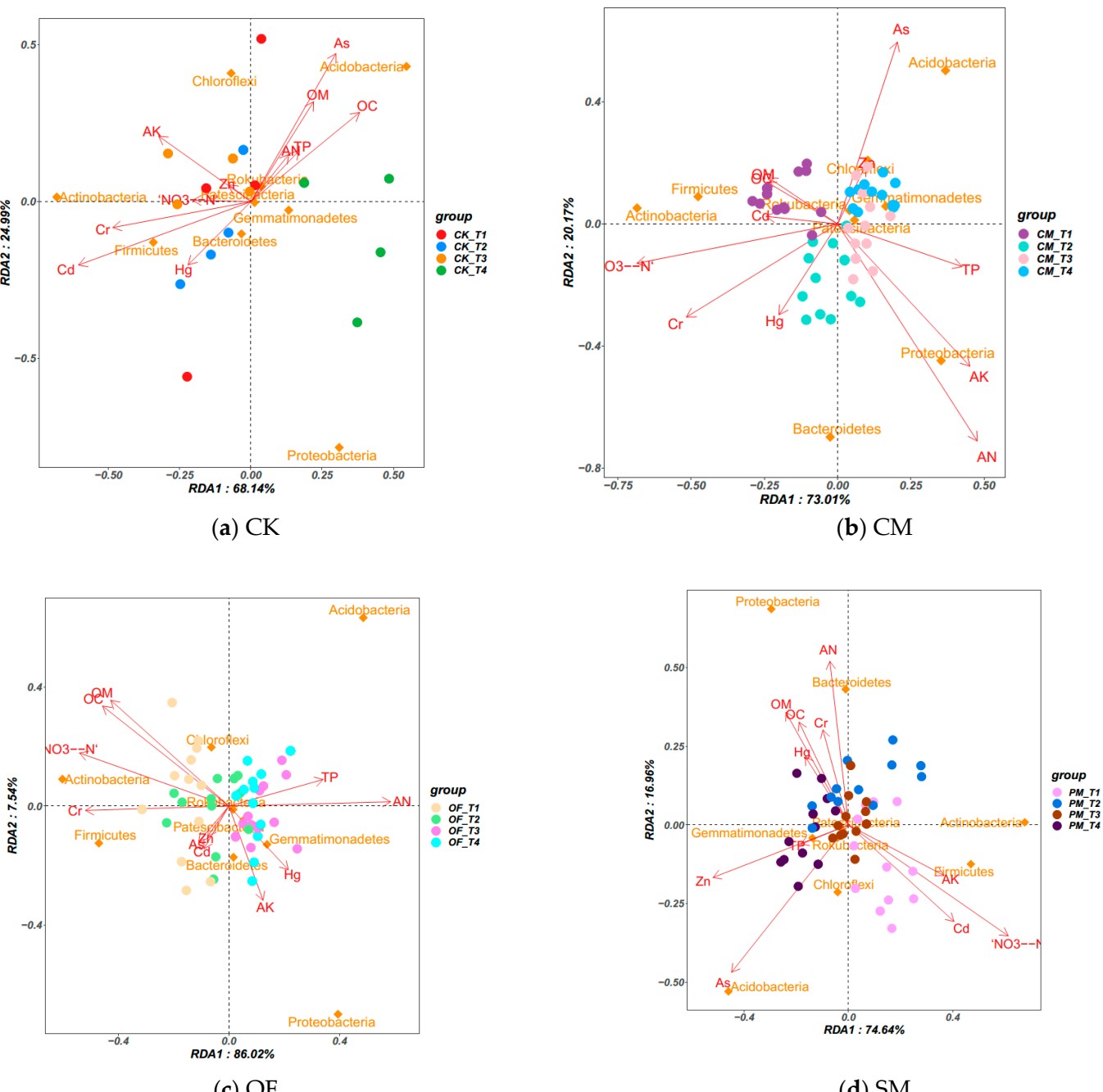

**Figure 5.** The relationship of environmental factors and microorganisms with different fertilizers (CK: control; CM: chicken manure; OF: organic fertilizer; PM: swine manure; T1: seedling period; T2: three-leaf period; T3: filling period; T4: mature period; OM: organic matter; OC: organic carbon; AN: available nitrogen; AK: available potassium; TP: total phosphorus).

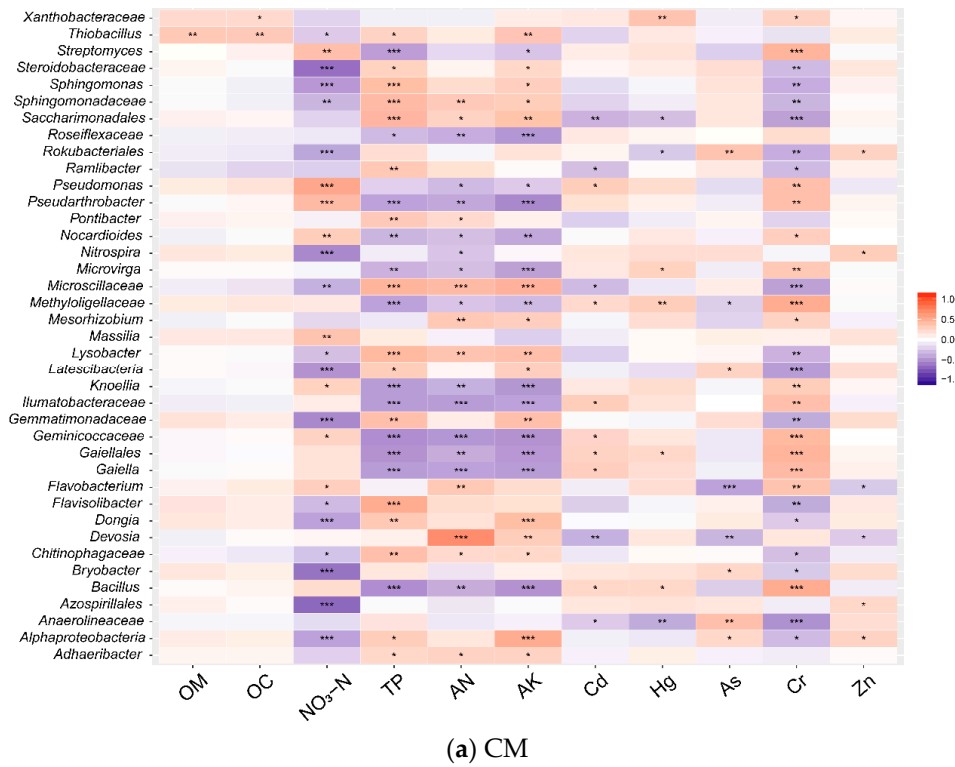

(**a**) CM

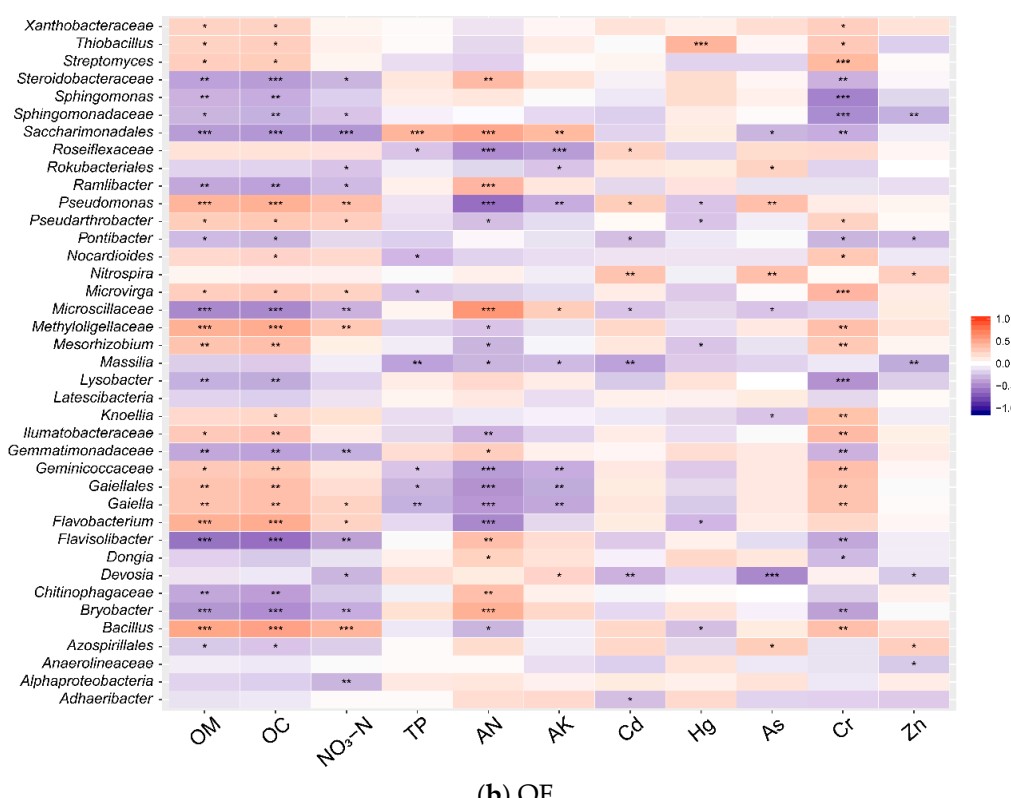

(**b**) OF

**Figure 6.** *Cont.*

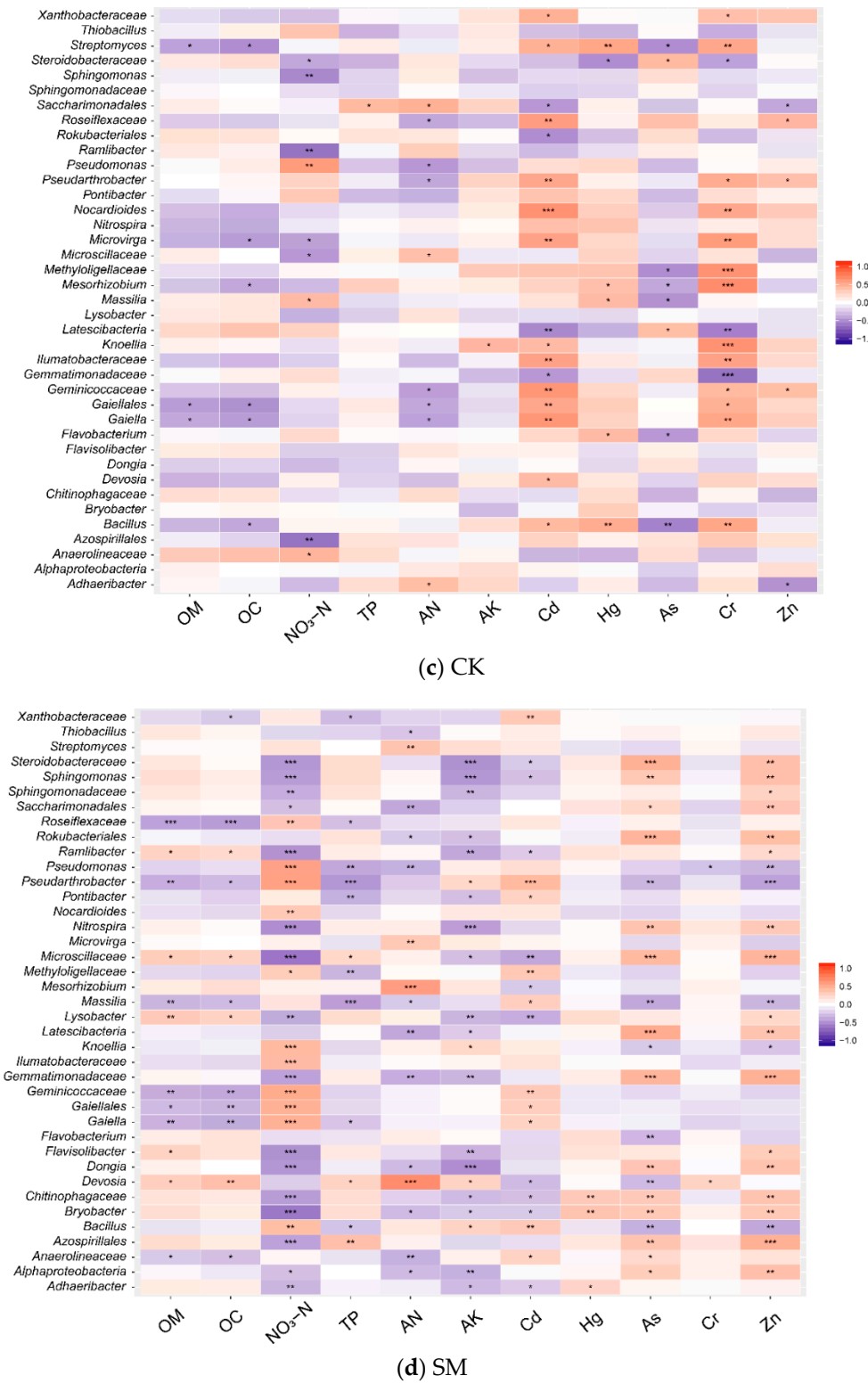

**Figure 6.** The Pearson correlation analysis of environmental factors and main genus of the CM (**a**), OF (**b**), control (**c**), and SM (**d**) tests (CM: chicken manure; OF: organic fertilizer; OM: organic matter; OC: organic carbon; TP: total phosphorus; AN: available nitrogen; AK: available potassium.* indicateds (*p* < 0.1), ** indicateds (*p* < 0.05),*** indicateds (*p* < 0.01)).

## 4. Conclusions

This study demonstrated that the microbial diversity varied during long-term fertilization, and the chicken manure was the best fertilizer to maintain the abundance of microorganisms. The microbial community of soil changes over time regardless of the addition of different fertilizers. Proteobacteria was the most abundant phylum in all the soils with different treatments. The second dominant phylum of each treatment group was Actinobacteria at the seedling and three-leaf stage, while the abundance of Acidobacteria was greatly increased with the chicken manure and organic fertilizer, becoming the second dominant phylum in the filling and mature periods. The applications of different fertilizers all increased the relative abundances of Firmicutes and Bacteroidetes. The correlations between environmental factors and microbial communities reflected that nutrient substances such as available nitrogen, available potassium, and nitrate became the most significant characteristics in the chicken and swine manure tests, while organic matter and ammonia exhibited similar effects on the microbial structure as with the organic fertilizer. The Pearson correlations of environmental factors on genus in the organic fertilizer tests were significantly different compared with the other tests. *Pseudomonas*, *Methyloligellaceae*, *Flavobacterium*, and *Bacillus* exhibited significant correlations with the organic matter. These results indicated that manure and organic fertilization directly affected soil bacterial diversity and community composition. However, the response mechanism of soil bacteria to organic fertilizer application is complex and needs further study to explain it satisfactorily.

**Author Contributions:** Conceptualization, C.S. (Chenyan Sha) and M.W.; methodology, C.S. (Chenyan Sha), J.W. (Jian Wu) and C.Y.; software, C.Y., J.W. (Jian Wu) and C.S. (Cheng Shen); validation, C.S. (Chenyan Sha), J.W. (Jian Wu) and J.S.; formal analysis, C.Y. and C.S. (Cheng Shen); investigation, J.W. (Jian Wu), C.Y. and C.S. (Cheng Shen); resources, M.W.; data curation, C.S. (Chenyan Sha) and J.W. (Jianqiang Wu); writing—original draft preparation, C.S. (Chenyan Sha) and J.W. (Jian Wu); writing—review and editing, C.S. (Chenyan Sha) and J.W (Jian Wu); visualization, C.S. (Chenyan Sha), J.W. (Jian Wu) and J.S.; supervision, C.S. (Chenyan Sha) and J.W. (Jian Wu); project administration, M.W.; funding acquisition, C.S. (Chenyan Sha). All authors have read and agreed to the published version of the manuscript.

**Funding:** This research was funded by the National Natural Science Foundation of China (grant numbers 31100404).

**Institutional Review Board Statement:** Not applicable.

**Informed Consent Statement:** Not applicable.

**Data Availability Statement:** Data can be found within the article.

**Acknowledgments:** This work was supported by the National Natural Science Foundation of China (grant numbers 31100404).

**Conflicts of Interest:** The authors declare no conflict of interest.

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
