# Peer review of "Effects of Different Fertilizers on Soil Microbial Diversity during Long-Term Fertilization of a Corn Field in Shanghai, China"

_diversity, doi:10.3390/d15010078_

Round 1
Reviewer 1 Report
The manuscript studies the influence of three different fertilizers on the nutrient content and microbial diversity of maize plants. The study was done in China.
The abstract is very well organized according to the Instructions to the authors.
The Introduction describes recent findings in the literature related to the manuscript.
Section Materials and methods are well developed, detailed, and clearly explained, including the experimental site layout. The statistical analysis and graphing were conducted using Excel 2019, SPSS 24 and Origin 2017. The differences were revealed through a one-way analysis of variance and with the LSD for multiple comparisons.
The Results start with a detailed analysis of physical and chemical characteristics, followed by Shannon and Ace diversity and richness indices of microbial community development under different fertilization schemes. In addition, the structure and microbial composition were analyzed through a metagenomic approach. The result was presented by a percentage of microbial abundance on phylum and genus levels and principal component analysis. Finally was assessed the correlation between the environmental factors and microbial communities was first analyzing RDA redundancy and Pearson correlation coefficient and, in the end, the Pearson correlation analysis of environmental factors and main genus.
The conclusions followed the findings and discussion.
The literature references comprised 32 sources, of which 38% are from the last five years.
Finally, I understand that the manuscript is very well written and easy to understand.
Author Response
Dear Professor and Editor
First, I would like to thank you for the helpful comments provided for our paper. In the paper we’ve addressed all comments. Besides, we improved the novelty and importance of this study and rewrote some parts of this paper carefully. And we also checked the language, grammar, punctuation, and spelling and overall style as well.
Reviewer 2 Report
Dear authors, the manuscript "Effect of different fertilizer on soil microbial diversity during long-term fertilization of a corn field in Shanghai, China" present an interesting and important topic for agronomy. The long-term experiments are of a major importance for a better understanding of microbial reaction to inputs and future forecasts.
There are some suggestions I consider will improve your work.
The Introduction present well the background of your research. I suggest you to modify the last paragraph to present clear the Aim of your research and the proposed Objectives. Each objective/hypotheses can be presented in a separate sentence.
Material and Methods section - please add in sub-section 2.1 Experimental location - the year when the field was established. This will help to explain the "long-term" from the title. Also, add the year of experiment. This is necessary to establish a time frame from the beginning to sampling.
Results and discussion section - This is a standard? "NY525-2012"- line 206. If yes, please add the reference.
You present some of nutrients/heavy metals as results. Please find a way to incorporate them into title.
The graphs from figure 2 were built with excel, spss or origin? Insert them into Statistical analysis sub-section. They look very nice and present well your data. Same observation for figure 3. This methodological approach is good for other studies to select the best visual expression of their results. Same observation for PCA (figure 4), RDA (Redundancy Analysis) in figure 5 and correlograms from figure 6. Expand the interpretation for figure 6. You have two pages of this figure and only one paragraph of interpretation. Add more comparison references for discuss in the Results and Discussion section. In this form you present well your results, but lack of comparison with other researches.
Overall, I like the idea of your research and I consider the manuscript deserve to be improved.
Author Response
Dear Professor and Editor
First, I would like to thank you for the helpful comments provided for our paper. In the paper we’ve addressed all comments. Besides, we improved the novelty and importance of this study and rewrote some parts of this paper carefully. And we also checked the language, grammar, punctuation, and spelling and overall style as well.
We provided an itemized reply to each of the reviewers' comments as follows:
For Reviewer #2
- Modify the last paragraph to present clear the aim of your research and the proposed Objectives.
Answer: We have modified the last paragraph to present clear the aim of our research and the proposed objectives by adding such sentences:These results indicated that manure and organic fertilization directly affected soil bacterial diversity and community composition. However, the response mechanism of soil bacteria to organic fertilizer application is complex and needs further study to explain it satisfactorily.
- Material and Methods section - please add in sub-section 2.1 Experimental location - the year when the field was established. Also, add the year of experiment.
Answer: We have already clarified the year when the field was established: The experimental base was established in 2010, which grows representative crops in Chongming District, mainly yellow corn and cauliflower. We also add the year of experiment in sub-section 2.3 Soil sample collection: Soil samples were collected in late April (seedling period), mid-May (three-leaf period), early July (filling period) and August Late (mature period) in 2020.
- Results and discussion section - This is a standard? "NY525-2012"- line 206. If yes, please add the reference.
Answer: We added the reference about the standard "NY525-2012".
- You present some of nutrients/heavy metals as results. Please find a way to incorporate them into title.
Answer: Since this paper is mainly about the community structure of soil microbial diversity, nutrients and heavy metals play auxiliary roles as physical and chemical properties.
- The graphs from figure 2 were built with excel, spss or origin? Insert them into Statistical analysis sub-section.
Answer: The graphs from figure 2 were built with origin and we inserted the sentence into Statistical analysis sub-section.
- Expand the interpretation for figure 6. You have two pages of this figure and only one paragraph of interpretation. Add more comparison references for discuss in the Results and Discussion section. In this form you present well your results, but lack of comparison with other researches.
Answer: We added more references to support the discussion to the figure 6
Reviewer 3 Report
Abstract demands to be slightly rewritten. Authors mentioned that they applied molecular biology technology, whereas nothing is known about methods used for nutrient contents and several heavy metals.
Line 59: should be: oligotrophic Acidobacteria
Please pay an attention that according the newest taxonomic rules only names of genera and species should be written in italic – please correct it in the whole ms text
Line 67: please add proper citation for confirmation
Line 81: you mean plants? by writing flora? or microorganisms? then should be biota…
Lines 117-119; 129-131 and later in the ms text: as you decide earlier to use abbreviations as CM, SF and OF please use them consequently instead of write i.e. chicken manure etc.
Line 144: please clarify the method of sampling, was the soil taken from a given point from one place or from several and mixed?
Lines 173-174: pay an attention on the proper font size
Line 173: should be indices instead of indexes
There is lack of information about PCR conditions, quality and quantity of DNA extracted as well as about method of sequencing and bioinformatic analyses – these aspects have to be clarified and supplemented
Line 202: what does it means calculated ± value? SE or SD? this should be pointed in the Table 1 title (the same is in Table 2)
Line 213: the highest
Lines 225, 226, 238: indices instead of indexes
Line 250: The Shannon…indices…
Line 283: Quality of Figure 3 demands to be improved, the legend in the current version is illegible
Line 367: if possible try to improve the quality of Figure 6 (genera names)
Author Response
Dear Professor and Editor
First, I would like to thank you for the helpful comments provided for our paper. In the paper we’ve addressed all comments. Besides, we improved the novelty and importance of this study and rewrote some parts of this paper carefully. And we also checked the language, grammar, punctuation, and spelling and overall style as well.
We provided an itemized reply to each of the reviewers' comments as follows:
For Reviewer #3
- Line 59: should be: oligotrophic Acidobacteria.
Answer: We changed “oligotrophic bacteria Acidobacteria” into oligotrophic Acidobacteria.
- Please pay an attention that according the newest taxonomic rules only names of genera and species should be written in italic -please correct it in the whole ms text.
Answer: We corrected it and all names of genera and species were written in italic.
- Line 67: please add proper citation for confirmation.
Answer: We checked the citation of Line 67: Dai, Z.; Su, W.; Chen, H.; Barberán, A.; Xu, J., Long-term nitrogen fertilization decreases bacterial diversity and favors the growth of Actinobacteria and Proteobacteria in agro-ecosystems across the globe. Global Change Biol. 2018, 24 (8).
- Line 81: you mean plants? by writing flora?
Answer: We changed the word into plants.
- Lines 117-119; 129-131 and later in the ms text: as you decide earlier to use abbreviations as CM, SF and OF please use them consequently instead of write i.e. chicken manure etc.
Answer: We used abbreviations as CM, SF and OF in some parts of our ms text.
- Line 144: please clarify the method of sampling, was the soil taken from a given point from one place or from several and mixed?
Answer: We clarified the method of sampling: According to the "S" sampling method, five points were randomly selected from each sample plot and rhizosphere soil samples were collected from each of the 5 points by using a sterile stainless steel soil drill (0-20 cm depth). Finally, the samples of five points will be mixed evenly.
- Lines 173-174: pay an attention on the proper font size.
Answer: We corrected the front size of Lines 173-174.
- Line 173: should be indices instead of indexes.
Answer: We changed the words “indexes” into “indices” .
- There is lack of information about PCR conditions, quality and quantity of DNA extracted as well as about method of sequencing and bioinformatic analyses- these aspects have to be clarified and supplemented.
Answer: We added the information about PCR conditions, quality and quantity of DNA extracted as well as about method of sequencing and bioinformatic analyses.
- Line 202: what does it means calculated ± value? SE or SD? this should be pointed in the Table 1 title (the same is in Table 2).
Answer: We used SD to represent the meaning of calculated ± value.
- Line 213: the highest
Answer: We corrected it.
- Lines 225, 226, 238: indices instead of indexes and Line 250: The Shannon…indices…
Answer: We corrected the words “indexes” and used indices.
- Quality of Figure 3 demands to be improved, the legend in the current version is illegible and Line 367: if possible try to improve the quality of Figure 6 (genera names).
Answer: We improved the quality of Figure 3 and Figure 6.
Round 2
Reviewer 2 Report
Dear authors,
The manuscript was improved and multiple suggestions were incorporated.
This form presents better your work and results.
Reviewer 3 Report
Authors responsed on all of my comments. The quality of ms was improved. I am satisfied with the ms quality and I reccommend to accept this paper.